# Transformation of internal solitary waves at the edge of the ice cover

Kateryna Terletska[1,2], Vladimir Maderich[2], and Elena Tobisch[1]

[1]Institut für Analysis, Johannes Kepler Universität Altenberger Straße 69, 4040 Linz, Austria
[2]Institute of Mathematical Machine and System Problems, Glushkov av., 42, Kyiv 03187, Ukraine

**Correspondence:** Vladimir Maderich (vladmad@gmail.com)

**Abstract.** Internal wave-driven mixing is an important factor in the balance of heat and salt fluxes in the polar regions of the ocean. Transformation of internal waves at the edge of the ice cover can enhance the mixing and melting of ice in the Arctic Ocean and Antarctica. In the Polar Oceans internal solitary waves (ISWs) are generated by various sources, including tidal currents over bottom topography, the interaction of ice keels with tides, time-varying winds, vortices, and lee waves. In this study, a numerical investigation of the transformation of ISW propagating from open water in the stratified sea under the edge of the ice cover is carried out to compare the depression ISW transformation and loss of energy on smooth ice surfaces, including those on the ice shelf and glacier outlets, with the processes beneath the ridged underside of the ice. They were carried out using a nonhydrostatic model, that is based on the Reynolds averaged Navier-Stokes equations in the Boussinesq approximation for a continuously stratified fluid. The Smagorinsky turbulence model extended for stratified fluid was used to describe the small-scale turbulent mixing explicitly. Two series of numerical experiments were carried out in an idealized 2D setup. The first series aimed to study the processes of the ISW of depression transformation under an ice cover of constant submerged ice thickness. Energy loss was estimated based on a budget of depth-integrated pseudoenergy before and after the wave transformation. The transformation of ISW of depression is controlled by the blocking parameter $\beta$ which is the ratio of the minimum thickness of the upper layer under the ice cover to the incident wave amplitude. The energy loss was relatively small for large positive and large negative values of $\beta$. The maximal value of energy loss was about $38\%$, and it was reached at $\beta \approx 0$ for ISW. In the second series of experiments, a number of keels were located on the underside of the constant thickness ice layer. The ISW transformation under ridged ice also depends on the blocking parameter $\beta$. For large keels ($\beta < 0$), more than $40\%$ of energy is lost on the first keel, while for relatively small keels ($\beta > 0.3$), the losses on the first keel are less than $6\%$. Energy losses due to all keels depend on the distance between them, which is characterized by the parameter $\mu$ which is the ratio of keel depth to the distance between keels. In turn, for a finite length of the ice layer the distance between keels depends on the keel quantity.

## 1 Introduction

Internal wave-driven mixing is an important factor in the balance of heat and salt fluxes in the polar regions of the ocean (Guthrie et al., 2013). In these areas, internal gravity waves are generated by various sources, including tidal currents over the bottom topography e.g. (Urbancic et al., 2022), time-varying winds (Rainville and Woodgate, 2009), vortices (Johannessen et al., 2019), and lee waves (Vlasenko et al., 2003). Another source of energy for internal waves in the near-surface pycnocline

can be an interaction of ice keels with tides (Zhang et al., 2022a). These waves, in the form of internal solitary wave (ISW), often propagate along the pycnocline in a stratified ocean under ice cover. The interaction between internal waves and ice cover is complex and depends on both the characteristics of the ice and the characteristics of internal waves (Carr et al., 2019). The

transformation of an ISW under an ice keel can cause the advection of water below the ice layer due to wave motion, whereas ISW shear and convective instabilities result in turbulent mixing. The heat advection and turbulent flux both will contribute to the vertical heat flux and consequently the change in temperature under the sea ice and increase of melting (Zhang et al., 2022b). An increased level of dissipation of the energy of internal waves propagating from the open water should be expected at the edge of the ice cover, which can represent the edge of an ice shelf or pack ice. In turn, the relief of the underside of

the ice and, in particular, the presence of ice keels can essentially affect ISW transformation, breaking, and energy dissipation. These aspects of the complicated problem of the interaction of internal waves and ice cover have not yet been investigated due to severe conditions for field observations in the polar regions of the ocean.

The problem of the transformation of a depression ISW under smooth ice cover is mathematically close to the problem of the transformation of an elevation IWS over a bottom step of constant height which has been considered analytically (Grimshaw

et al., 2008) and numerically using a nonhydrostatic model (Maderich et al., 2009; Talipova et al., 2013). It was found that the transformation of an ISW over the step in a two-layer fluid depends on the ratio of the thickness of the lower layer over the step to the ISW amplitude. The transformation of the elevation ISW over a single obstacle (ridge) on the bottom has been studied in the laboratory (Wessels and Hutter, 1996; Chen, 2007; Du et al., 2021) and numerically (Vlasenko and Hutter, 2001; Xu et al., 2016). Wave breaking on the lee side of the ridge was accompanied by the generation of second mode ISWs. The propagation

of an elevation ISW over a corrugated bed (Carr et al., 2010) was accompanied by shear instability in the form of billows. ISWs propagating from open water to ice were studied in the laboratory by Carr et al. (2019) for grease, level, and nilas ice. The experiments showed that the dissipation of turbulent kinetic energy under the ice is comparable to that of the ISW in the water column. The disintegration of an ISW of depression under a single ice keel was simulated by Zhang et al. (2022b). It was concluded that corresponding turbulent mixing can enhance the melting of ice keels.

In this study, a numerical investigation of the transformation of an ISW propagating from ice-free water in the stratified sea under the edge of the ice cover is carried out to compare the depression ISW transformation and loss of energy on smooth ice surfaces, including those on the ice shelf, with the processes beneath the ridged underside of the ice. The rest of the paper is organized as follows. The formulation of the problem, the model setup, and the relevant numerical tools are given in Section 2. Section 3.1 presents the simulation results for smooth ice cover, whereas the results of the simulation of ridged ice cover are

considered in Section 3.2. The results of the simulations are summarized and discussed in Section 4.

## 2 Numerical experiment setup

The numerical simulations were carried out using a nonhydrostatic model (Maderich et al., 2012). The numerical model used here is based on the Reynolds averaged Navier–Stokes equations in the Boussinesq approximation for a continuously stratified fluid. The Smagorinsky turbulence model extended for stratified fluid (Siegel and Domaradzki, 1994) was used to explicitly

describe the small-scale turbulent mixing in the ocean-scale ISWs. Two series of numerical experiments were carried out in an idealized 2D setup. The first series aimed to study processes of the ISW of depression transformation under ice cover of constant submerged ice thickness (draft) $h_{ice}$ (Fig. 1a). The second series was carried out to simulate the effect of ridged ice on ISW of depression propagation in a similar two-layer stratification (Fig. 1b). A computational tank of constant depth $H = 200$ m and length $L = 10000$ m was used. It was assumed that the ice layer of length $L_{ice} = 5000$ m is rigid and does not interact with the ISWs. The coordinate $x$ is directed along the computational domain, and $z$ is directed vertically upward. Idealized stratification of the vertical distribution of potential density is considered in the form:

$$\sigma_\theta = \frac{(\sigma_{\theta 2} - \sigma_{\theta 1})}{2} \tanh\left(\frac{z - h_1}{\Delta h}\right) + \frac{(\sigma_{\theta 2} + \sigma_{\theta 1})}{2}, \tag{1}$$

where $h_1$ is the thickness of the upper layer of water in the absence of the ice cover, $h_2 = H - h_1$ is the thickness of the lower layer, whereas $h_{1+} = h_1 - h_{ice}$. As shown by Maderich et al. (2010) and Talipova et al. (2013), the transformation of both elevation and depression ISWs is controlled by the blocking parameter $\beta$ which is the ratio of the height of the minimum thickness of the upper layer under the ice cover $h_{1+}$ to the incident wave amplitude $a_i$:

$$\beta = \frac{h_{1+}}{a_i}, \tag{2}$$

where $\beta$ is positive in the case $h_{1+} > 0$, $(h_1 > h_{ice})$ and negative for $h_{1+} < 0$, $(h_1 < h_{ice})$. In the first series submerged ice thickness $h_{ice}$ was constant along the computational tank varying in different numerical experiments from $0.5$ m to $40$ m (Table 1). In the second series of experiments, several keels were placed underside of the ice layer of constant thickness $h_{ice}$ (Fig.1b). The ice keel shape was approximated by Versoria function (Skyllingstad et al., 2003) as

$$h_{keel}(\delta x) = \frac{h_k b_k^2}{b_k^2 + (\delta x)^2}, \tag{3}$$

where $h_k$ is maximal keel penetration, $b_k$ is a parameter governing to determine the keel width, $\delta x = x - x_k$ is the horizontal distance from the centre of the keel placed at $x_k$. The keel form was similar: i.e. $h_k / b_k$ is constant. Following Zhang et al. (2022b), we define keel width as the horizontal width of the consolidated ice zone at a depth of $4$ m below the bottom of the ice (Marchenko, 2008). Typical values of $h_k$ are $3 - 28$ m (Strub–Klein and Sudom, 2012) reaching $45$ m (Leppäranta, 2007), whereas typical keel width varies in the range of $3 - 200$ m (Strub–Klein and Sudom, 2012). In the ocean, the ratio of the maximum height of the keel $h_k$ to the distance between the keels $L_k$ varies from $1/20$ for heavily ridged ice to $1/1000$ for moderately ridged ice (Lu et al., 2011). For this idealized case study, the vertical distribution of potential density anomaly mimics the summer profile of potential density over the Yermak Plateau (Randelhoff et al., 2017) in the Arctic Ocean (1), where $h_1 = 20$ m, $\triangle h = 10$ m, $\sigma_{\theta 1} = 25.4$ kg m$^{-3}$, $\sigma_{\theta 2} = 27.7$ kg m$^{-3}$. As seen in Fig. 2, the summer profile of density does not have a well-mixed surface layer due to the stratification caused by ice melting. Free-slip boundary conditions were used at all boundaries except along the ice-water boundary. The Neumann-type boundary condition for the nonhydrostatic pressure component was used at the solid boundaries. At the free surface and open boundaries, this component was set zero (Maderich et al., 2012). At the corner of the underwater step, this condition is violated. However, numerical experiments for different resolutions have shown that this problem does not occur at simulated fields of velocity and density. Ice-ocean tangential stress

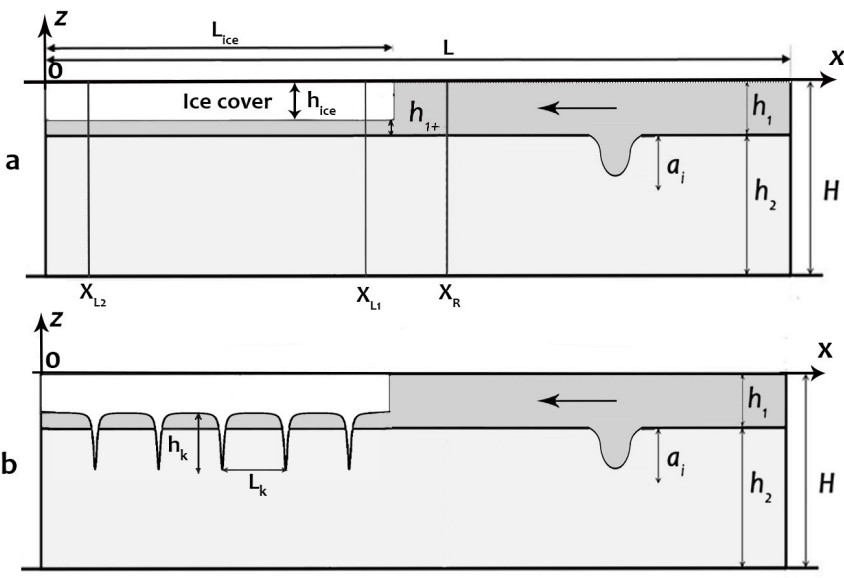

**Figure 1.** Sketch of the numerical configuration for simulation of ISWs transformation under the ice. (a) Smooth ice cover. (b) Ridged ice.

is parameterized using the quadratic bulk formula with a drag coefficient $C_D$. The value of $C_D$ under ice varies in the range $10^{-3} - 10^{-2}$ (Lu et al., 2011). No-flux condition was also used at all boundaries. The model was initialized using the iterative solution of the Dubreil-Jacotin-Long (DJL) equation (Dubreil-Jacotin, 1932) with the initial guess obtained from a weakly

nonlinear theory. The DJLES spectral solver from the MATLAB package https://github.com/ mdunphy/DJLES/ was used to generate ISW of depression. To get around the difficulties associated with the numerical solution of the nonhydrostatic model equations in the presence of an ice layer, we considered the setting mirrored for the upper surface of the ocean, in which the ice layer was replaced by a step on the bottom. Then the vertical profile (1) was replaced by the distribution

$$\sigma_\theta = \frac{(\sigma_{\theta 1} - \sigma_{\theta 2})}{2} \tanh\left(\frac{z - (H - h_1)}{\Delta h}\right) + \frac{(\sigma_{\theta 2} + \sigma_{\theta 1})}{2}, \tag{4}$$

where $\sigma_{\theta 1} = 27.7$ kg m$^{-3}$, $\sigma_{\theta 2} = 25.4$ kg m$^{-3}$. The initial ISW of depression was changed, respectively, to a ISW of elevation. This approach is accurate when we consider the problem with rigid lid approximation at the free surface. However, the numerical model is a free-surface model, which leads to bottom fluctuations outside the step in the computational flume. Therefore, we conducted tests with ISWs of the same amplitude propagating as an ISW of depression and as an ISW of elevation in stratification (1) and (4). The tests aimed to estimate the effect of the free surface on the wave characteristics for

free-slip boundary conditions. These results demonstrate a weak effect of the free surface on ISW dynamics in the considered cases, which made it possible in this problem to replace the conditions on the free surface with conditions on the rigid lid. The results of the comparison for horizontal velocity taking into account the mirror reflection of the vertical coordinate in Fig. 3 showed that the difference in the velocity between the two configurations of the model does not exceed 1%. Note that in

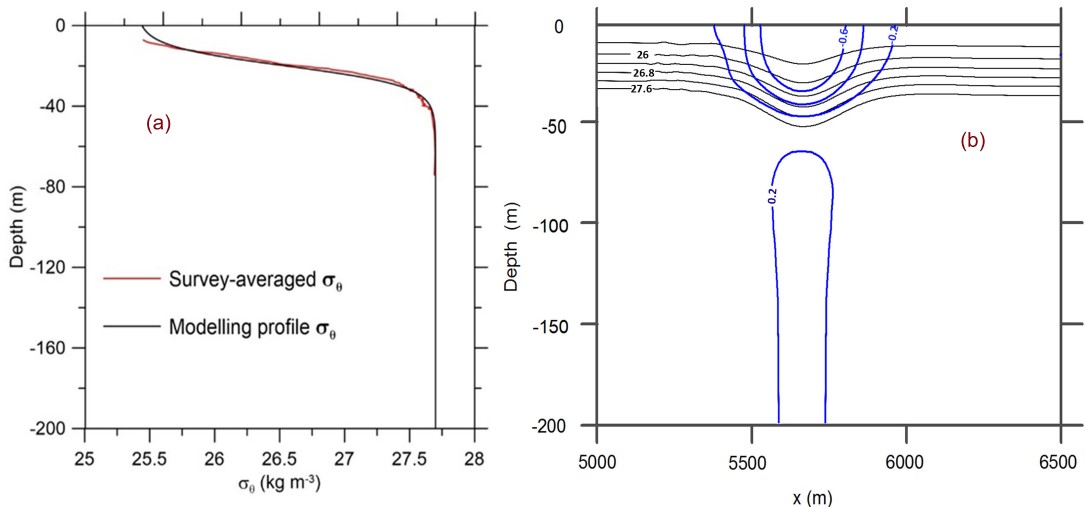

**Figure 2.** The comparison of the background stratification in the computational tank (1) with the survey averaged profile of anomaly of potential density $\sigma_\theta$ (Randelhoff et al., 2017) (a); The vertical cross-section of potential density and horizontal velocity fields in the incident ISW of amplitude 15 m (b).

laboratory experiments (Carr et al., 2008; Luzzatto-Fegiz and Helfrich, 2014) the influence of a free surface on the stability
of waves with a trapped core was shown. This effect has been interpreted as the influence of surfactants which are essential in laboratory-scale processes. However, these Marangoni effects have a negligible impact on the interior of full-scale oceanic waves (Luzzatto-Fegiz and Helfrich, 2014). In the first series of experiments, 48 runs were performed using the generalized

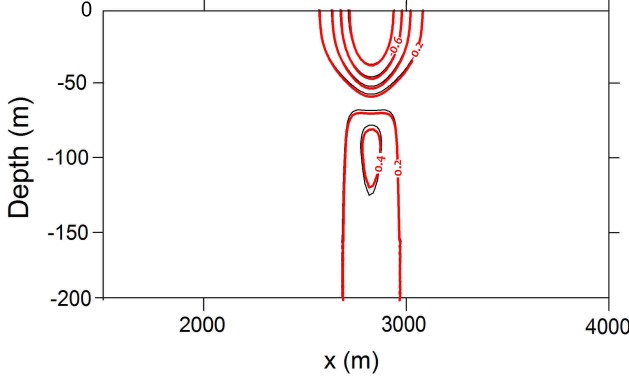

**Figure 3.** The comparison of the vertical cross-section of horizontal velocity fields in a wave of depression (black line) and a wave of elevation (red line) in stratification (1) and (4) respectively for ISW with an amplitude 33 m.

**Table 1.** The parameters of the first series of runs.

| Run | $a_i$ | $h_{ice}$ | $a_i/h_1$ | $C_D$ | $\beta$ |
|-----|-------|-----------|-----------|-------|---------|
| | m | m | | | |
| 1-6 | 33 | 0.5, 5, 10, 20, 30, 40 | 1.65 | 0.001 | $0.6, 0.45, 0.3, 0, -0.3, -0.6$ |
| 7-12 | 25 | 0.5, 5, 10, 20, 30, 40 | 1.25 | 0.001 | $0.78, 0.6, 0.4, 0, -0.4, -0.8$ |
| 13-18 | 15 | 0.5, 5, 10, 20, 30, 40 | 0.75 | 0.001 | $1.3, 1, 0.7, 0, -0.7, -1.33$ |
| 19-24 | 8 | 0.5, 5, 10, 20, 30, 40 | 0.4 | 0.001 | $2.44, 1.88, 1.25, 0, -1.25, -2.5$ |
| 25-30 | 33 | 0.5, 5, 10, 20, 30, 40 | 1.65 | 0.01 | $0.6, 0.45, 0.3, 0, -0.3, -0.6$ |
| 31-36 | 25 | 0.5, 5, 10, 20, 30, 40 | 1.25 | 0.01 | $0.78, 0.6, 0.4, 0, -0.4, -0.8$ |
| 37-42 | 15 | 0.5, 5, 10, 20, 30, 40 | 0.75 | 0.01 | $1.3, 1, 0.7, 0, -0.7, -1.33$ |
| 43-48 | 8 | 0.5, 5, 10, 20, 30, 40 | 0.4 | 0.001 | $2.44, 1.88, 1.25, 0, -1.25, -2.5$ |

vertical system of coordinates (Maderich et al., 2012). The vertical and horizontal grid resolution was $400 \times 3000$. The quasi-z-level coordinate system (Maderich et al., 2012) was used to describe this step-like ice layer. These runs cover a range of
incident ISWs with moderate, $a_i = 8$ m, and large amplitudes, $a_i = 33$ m (Table 1). The incident ISW amplitude is defined as the maximum displacement of the undisturbed isopycnals. The wavelength $\lambda_{0.5}$ is estimated as the half-width at the depth where the amplitude of the wave is reduced by half. Two cases with different drag coefficients ($C_D = 0.001$ and $C_D = 0.01$) were considered to investigate the influence of ice roughness on ISW transformation and energy loss. A wide range of ice cover drafts $h_{ice}$ from 0.5 m to 40 m were used to investigate processes under ice cover from first-year ice to the ice shelf front. In
the second series of experiments (see Table 2), 12 runs (K1-K12) were performed using a sigma-system of coordinates, which allowed for accurately describing flow around the keel. The vertical and horizontal grid resolution was also 400 x 3000. The density stratification in this series was the same as in the first series. The ISW amplitude was $a_i = 15$m, wavelength $\lambda_{0.5} = 320$ m, and drag coefficient $C_D = 0.001$. The ice draft was 1 m which mimics one-year ice. The keel's maximal penetration $h_k$ varied from 7 m to 21 m, and the width of the keel varied from 67 m to 200 m at a depth of 4 m. To take into account the
geometric characteristics of the keels and their frequency, we introduce the parameter $\mu$ which is the ratio of sum of submerged ice thickness and maximal keel penetration to the distance between keels

$$\mu = (h_{ice} + h_k)/L_k. \tag{5}$$

In turn, for a finite length of the ice layer $L_{ice}$ the distance between keels depends on keel number $n$ as $L_k = L_{ice}/n$. In the second series $\mu$ varies in experiments $K1 - K3$ from 0.008 to 0.088 (Table 2). To directly compare with the step case,
calculations were made for step (S1) with the same draft as the height of the keel in the K1-K4 experiments. In another experiment (S2), the step draft was chosen to be equal to the average draft in experiment K2.

**Table 2.** The parameters of the second series of runs

| Run | $h_{ice}$ | $h_k$ | $b_k$ | $L_k$ | $\beta$ | $\mu$ | $E_{loss}$ | $E_{tot}$ |
|-----|-----|-----|-----|-----|-----|-----|-----|-----|
|  | m | m | m | m |  |  | % | % |
| $K1$ | 1 | 21 | 49.5 | 250 | $-0.13$ | 0.088 | - | 82.4 |
| $K2$ | 1 | 21 | 49.5 | 500 | $-0.13$ | 0.044 | - | 76.3 |
| $K3$ | 1 | 21 | 49.5 | 1000 | $-0.13$ | 0.022 | - | 64.3 |
| $K4$ | 1 | 21 | 49.5 | $>5000$ | $-0.13$ | – | 41.2 | 47.4 |
| $K5$ | 1 | 14 | 33 | 250 | 0.33 | 0.06 | - | 42.6 |
| $K6$ | 1 | 14 | 33 | 500 | 0.33 | 0.03 | - | 40.2 |
| $K7$ | 1 | 14 | 33 | 1000 | 0.33 | 0.015 | - | 29.8 |
| $K8$ | 1 | 14 | 33 | $>5000$ | 0.33 | – | 6.3 | 13.2 |
| $K9$ | 1 | 7 | 16.5 | 250 | 0.8 | 0.032 | - | 43.6 |
| $K10$ | 1 | 7 | 16.5 | 500 | 0.8 | 0.016 | - | 37.3 |
| $K11$ | 1 | 7 | 16.5 | 1000 | 0.8 | 0.008 | - | 28.6 |
| $K12$ | 1 | 7 | 16.5 | $>5000$ | 0.8 | – | 3.5 | 10.2 |
| $S1$ | 22 | 0 | – | – | $-0.13$ | – | 36.2 | 75.2 |
| $S2$ | 5 | 0 | – | – | 1 | – | 8.8 | 22 |

## 3 Results

### 3.1 First series of experiments

The results of the first series of experiments for the transformation of a depression ISW for an incident wave at the ice were
given for a wide range of ice drafts, incident wave amplitudes, and drag coefficients (Table 1). The snapshots of the density field
for an incident ISW of amplitude $a_i = 15$ m passing under the ice cover are shown in Fig. 4 for different $\beta$. Transformation
under thin ice ($h_{ice} = 0.5$ m) with $\beta = 1.3$ occurs without any instability and essential disturbances. For increased ice draft
$\beta = 0.7$ ($h_{ice} = 10$ m), the incident wave changes its form and amplitude as it passes under the ice. The amplitudes of reflected
and transmitted waves were well predicted by the theoretical model (Grimshaw et al., 2008). For $\beta = 0$ ($h_{ice} = 20$ m) the
transmitted wave has a smaller amplitude, and more energy is transferred to the reflected wave at the ice edge. Waves under the
ice transform into strongly nonlinear boluses, and more energy goes to the reflected waves when the draft of the ice is equal
to the depth of the upper layer ($\beta = 0$). The bolus under the ice becomes smaller and reflected waves form as a result of the
strong interaction with the ice front at $\beta = -0.7$ ($h_{ice} = 30$m). An important characteristic of the ISW-ice interaction is the
loss of kinetic and available potential energy during the ISW transformation. Energy transformation due to mixing leads to the
transfer of energy to background potential energy and energy dissipation. An energy loss was estimated based on a budget of
depth-integrated pseudoenergy before and after the wave transformation following (Lamb, 2007) and (Maderich et al., 2010).
The characteristics of the incoming and reflected wave were recorded in the cross-sections $X_R$ (Fig. 1a), which are located

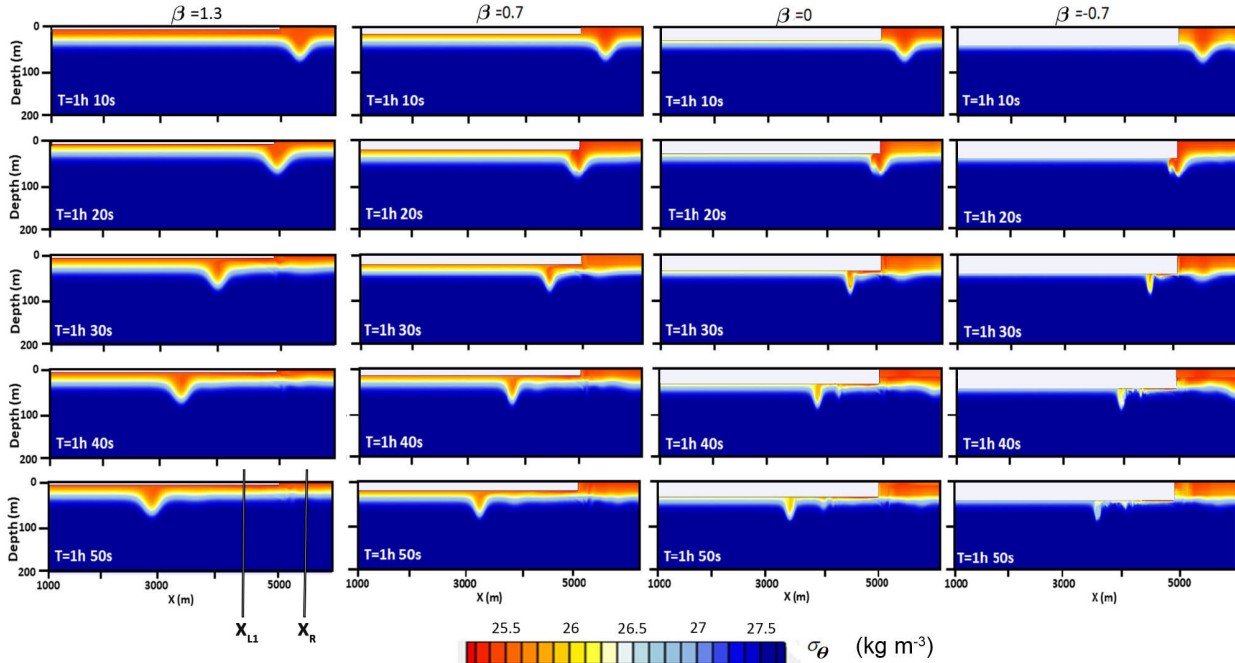

**Figure 4.** The snapshots of the density field for incident ISW with amplitude 15 m passing under the ice cover with a different draft. The integration region for the energetics calculations between $X_{L1}$ and $X_R$ is shown.

near the ice edge and in two cross-sections ($X_{L1}$ placed at a distance of 500 m from the ice edge, and $X_{L2}$ placed at a distance

of 4500 m from the ice edge (Fig. 1a). The energy loss $\Delta E_{loss}$ in the cross-section $x_{L1}$ characterizes energy transformations in the vicinity of ice edge, whereas energy losses $\Delta E_{visc}$ take into account the dissipation of energy due to the underside ice friction effects. The total energy of the incident, reflected, and transmitted waves was calculated using the depth-integrated pseudoenergy flux $F(x,t)$ to find the balance of the total energy

$$F(x,t) = \int_{-H}^{0} (E_{PSE} + p)U dz, \qquad (6)$$

where $p$ is the pressure disturbance due to the passing wave, $U$ is the horizontal velocity, and $E_{PSE}$ is the pseudoenergy

density, which is a sum of kinetic energy density $E_k$ and available potential density $E_a$ (part of the potential energy available for conversion into kinetic energy). For the calculation of $E_a$, we used a reference density profile that was obtained by an adiabatic rearranging of the density field. The volume integration of these flows outside the mixing zone allows us to estimate the energy of the incoming $PSE_{in}$, reflected $PSE_{ref}$, and transmitted under ice ISWs in cross-sections $X_{L1}$ and $X_{L2}$ (see

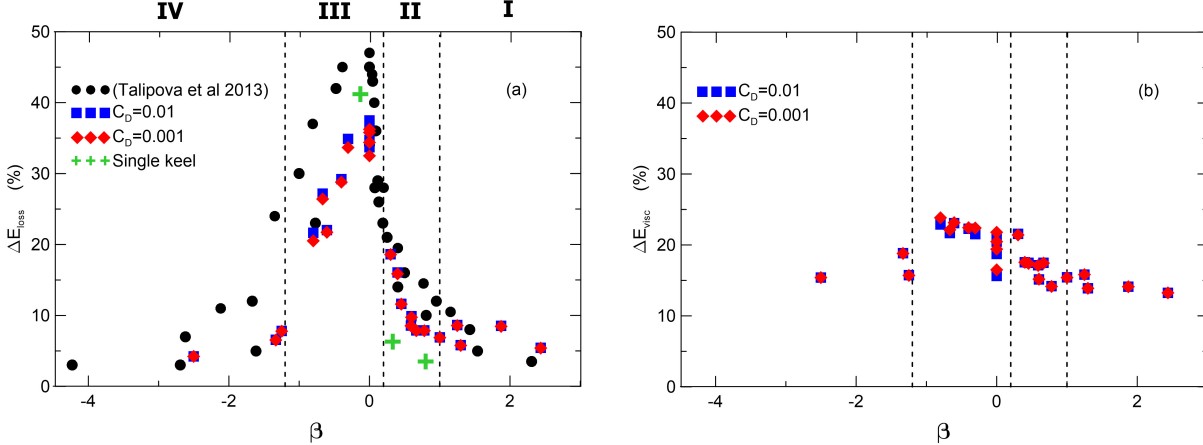

**Figure 5.** (a) The ISW energy loss $\Delta E_{loss}$ under the ice cover versus the blocking parameter $\beta$ for various amplitudes of an incident wave (b) $\Delta E_{visc}$ that take into account the dissipation of energy due to the underside ice friction effects versus the blocking parameter $\beta$.

Fig. 1 a )$PSE_{tr1}$ and $PSE_{tr2}$ respectively:

$$
\begin{aligned}
PSE_{in} &= \int_{X_R}^{L} \int_{-H}^{0} E_{PSE} dz dx = -\int_{t_1}^{t_2} F(X_R, t) dt, \\
PSE_{tr1} &= \int_{0}^{X_{L1}} \int_{-H}^{0} E_{PSE} dz dx = -\int_{t_2}^{t_3} F(X_{L1}, t) dt, \\
PSE_{ref} &= \int_{X_R}^{L} \int_{-H}^{0} E_{PSE} dz dx = \int_{t_2}^{t_3} F(X_R, t) dt \\
PSE_{tr2} &= \int_{0}^{X_{L2}} \int_{-H}^{0} E_{PSE} dz dx = -\int_{t_4}^{t_5} F(X_{L2}, t) dt,
\end{aligned}
\tag{7}
$$

where $t_2 - t_1$ is the interval of time when the incoming wave passes the cross-section $X_R$, $t_3 - t_2$ is the interval of time when transmitted and reflected wave pass the cross-sections $X_{L1}$ and $X_R$ respectively. Time interval $t_5 - t_4$ corresponds to the transmitted wave passes the cross-section $X_{L2}$. The normalized energy loss $\Delta E_{loss}$, $\Delta E_{visc}$ and $\Delta E_{tot}$ are given by

$$
\begin{aligned}
\Delta E_{loss} &= (PSE_{in} - PSE_{tr1} - PSE_{ref})/PSE_{in}, \\
\Delta E_{tot} &= (PSE_{in} - PSE_{tr2} - PSE_{ref})/PSE_{in}, \\
\Delta E_{visc} &= \Delta E_{tot} - \Delta E_{loss} = (PSE_{tr1} - PSE_{tr2})/PSE_{in}.
\end{aligned}
\tag{8}
$$

The energy loss as a result of ISW transformation under ice $\Delta E_{loss}$ at interval $X_R$–$X_{L1}$ versus the blocking parameter $\beta$ is shown in Fig. 5a. This loss was relatively small for large positive and large negative values of $\beta$. The maximal value of energy loss was about $38\%$, and it was reached at $\beta \approx 0$. The character of energy losses and the relationship between transmitted and reflected ISW energy allows us to distinguish different regimes for ISW interaction under ice cover: the weak interaction (I), moderate interaction (II), strong interaction (III), and reflection regime (IV). The weak interaction (I) is when the ISW transforms under ice cover without any instability; the energy losses are mainly due to viscous dissipation. It corresponds to values $\beta > 0.5$. The energy losses at cross-sections $X_R$ - $X_{L1}$ are about $10\%$. The amplitudes and numbers of reflected and transmitted waves are well predicted by the theoretical model of (Grimshaw et al., 2008). Moderate interaction (II) occurs

when the waves become unstable under ice cover, resulting in energy losses due to the turbulent mixing varying from $10\%$ to $20\%$. The strong interaction (III) of the ISW with the ice is the regime when the flow under the ice is supercritical. This regime is identified by the condition that the maximal composite Froude number $Fr_{max}$ at the step cross-section is greater than 1, where $Fr$ is defined as

$$Fr^2 = \frac{(U_1)^2}{g'h_1(x)} + \frac{(U_2)^2}{g'h_2},$$
(9)

where $U_1$ and $U_2$ are the layer-averaged velocities in each layer, $g' = g \cdot \Delta\rho/\rho_0$, where $g$ is the gravity acceleration, $\Delta\rho$ and $\rho_0$ are the density difference between upper and lower layers and undisturbed density of fluid, respectively. Supercritical flow $Fr_{max} = 1$ with $\beta = 0$ resulted in bolus formation and intensive mixing, which reached about $40\%$. The reflection regime (IV) is when the height of the ice floe is large enough to result in full reflection of the ISW. The energy losses are again small ($\triangle E_{loss}$ less than $10\%$–$15\%$). In this regime, energy losses depend on the wave amplitude; small and moderate incident waves reflect without turbulent mixing. This dependence of $\Delta E_{loss}$ on $\beta$ is comparable to values for a bottom step (Talipova et al., 2013) obtained using direct simulation by the Navier-Stokes equations (Fig.5a). The differences in values of the energy losses from (Talipova et al., 2013) and from the present investigation can be explained by the fact that the field scale problem was studied in this work using the Reynolds averaged equations, while in (Talipova et al., 2013) the propagation of ISWs in a laboratory-scale computational domain was studied by using the Navier-Stokes equations. The eddy viscosity and diffusivity calculated from the turbulence model (Siegel and Domaradzki, 1994) vary in space and time with characteristic values $10^{-4} - 10^{-3}$ m$^2$s$^{-1}$. The difference between the energy losses in the cross-sections $X_{L2}$ and $X_{L1}$ characterizes their losses due to friction effects. This difference $\Delta E_{visc} = \Delta E_{tot} - \Delta E_{loss}$ is shown in Fig.5b as the function of $\beta$. This shows that, the contribution of friction is $15 - 20 \%$ of the energy of the incident wave. The simulations showed a weak dependence of energy loss on the friction parameter $C_D$ (Fig. 5b).

## 3.2    Second series of experiments

The results of the second series of experiments for the transformation of depression ISWs under ridged ice for different ice keel heights and distances between keels (Table 2) are discussed in this section. Similar to (2), we can introduce the blocking parameter for a single keel in the form

$$\beta = \frac{h_1 - h_{ice} - h_k}{a_i}.$$
(10)

The snapshots of the density field for an incident ISW passing under the layer of constant draft (Run S1 from Table 2) are compared in Fig. 6 with the results for a wave passing under ridged ice (Run K2). In Run S1 the ISW amplitude is comparable to the draft of ice and thickness of the upper layer $h_{1+}$ therefore interaction was strong. Initially (time interval: $T = 1$ h 30 m - 1 h 35 m ), the wave propagated under the ice as a bolus (Fig. 6 a). This process is accompanied by intensive mixing. The bolus gradually loses mass. Estimates of energy loss at a distance of $500$ m from the ice front $E_{loss}$ in (Run S1 in Table 2) showed that $36.2\%$ of energy was lost to mixing and dissipation, whereas loss of energy at the full length of ice cover ($5000$ m) was twice as much ($E_{tot}$=75.2%). The processes of ISW disintegration and mixing for ridged ice differ essentially from the case

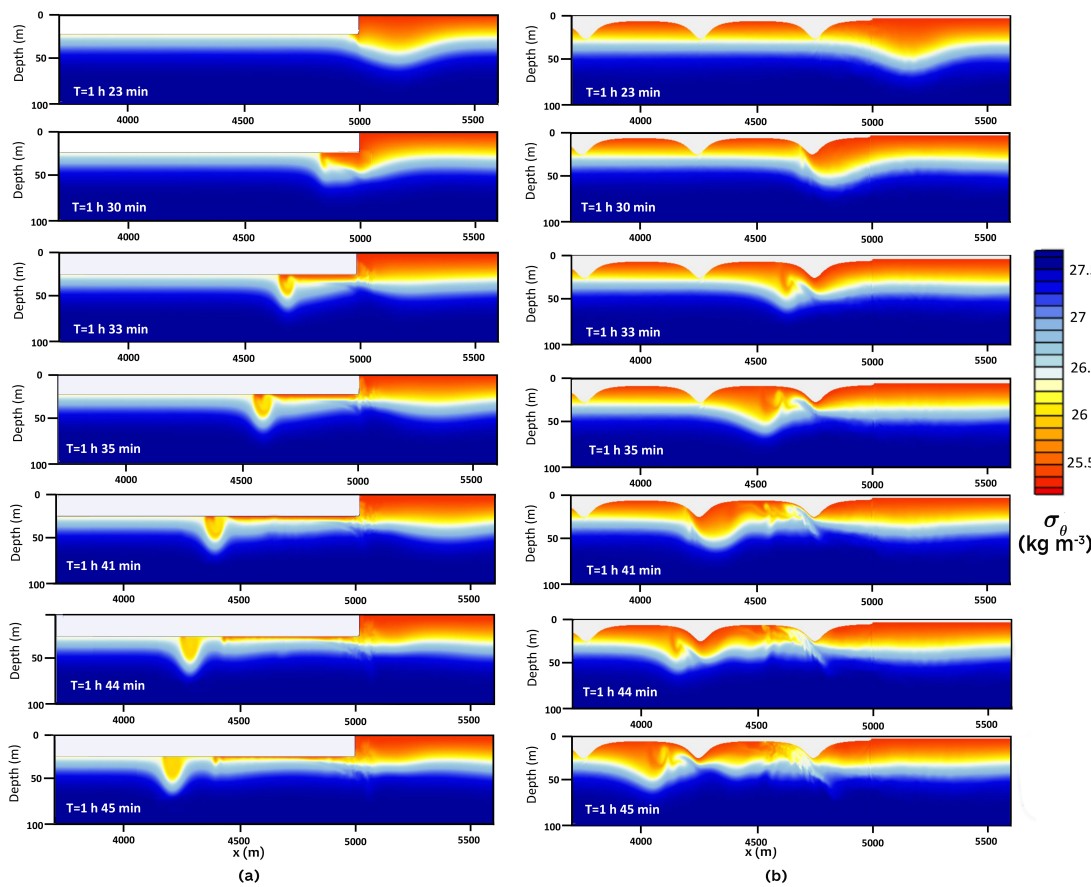

**Figure 6.** (a) The transformation of ISW of depression under a smooth ice layer (Run S1); (b) The transformation of ISW of depression under a ridged ice layer (Run K2).

of the ice layer of the constant draft. The snapshots of the density field for the ISW passing under the ridged ice (Run K2) are shown in Fig. 6 b. As seen in the figure, the flow accelerates at the rear side of the keel ( $T = 1$ h 30 m, $Fr_{max}$ reaches the value 1) entraining denser water from the underlying layers. The resulting vortex is accompanied by intense mixing ($T = 1$ h 33 m -1 h 45 m). The process of transformation of this wave with a slightly smaller amplitude is repeated on subsequent keels. As a result of passing through the first keel, the wave loses about $41\%$ of incident wave energy. Energy losses due to all keels depend on the distance $L_k$ between them, that in turn depend on keel quantity. When $\beta = -0.13$ then $E_{tot}$ changes from $47.4\%$ for a single keel to $82.4$ % for $L_k = 250$ m. This means that energy losses on the first keel account for about half of all losses. For $L_k = 1000$ m, the energy loss due to all keels was $64.3$ %. As $\beta$ increases to $0.8$, the contribution of the first keel decreases to $3.5$ %. In the limiting case of the interaction of ISW with a single keel (Zhang et al., 2022b), the maximum energy dissipation

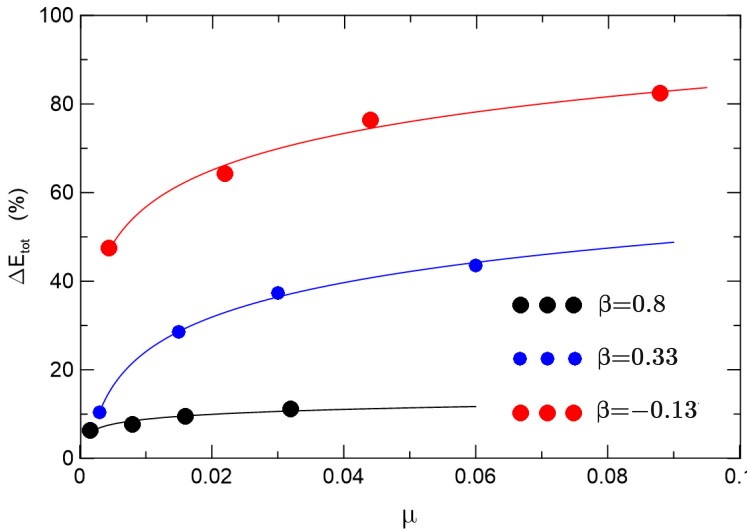

**Figure 7.** The ISW energy loss $\triangle E_{loss}$ under the ice cover versus parameter $\mu$. The logarithmic curves approximate calculated dependencies.

was about $25\%$ which is somewhat less than in our calculations, but we need to keep in mind the differences in the calculation
parameters and turbulence parameterization. Zhang et al. (2022b) used constant eddy coefficients whereas in our study the
turbulence model was used with eddy coefficients varying in space and time. To characterize the dependence of $\Delta E_{tot}$ on keel
height and distance between keels we introduced parameter $\mu$ (5). As seen in Fig. 7 this dependence can be approximated
by logarithmic curves $\Delta E_{loss} = q\ln\mu + r$, where $\beta = (0.8, 0.33, -0.13)$, $q = (1.60, 11.24, 11.96)$, $r = (16.21, 75.84, 111.85)$.
The energy loss $\Delta E_{tot}$ increases with the decrease of distance between keels or an increase of keel height. The level of $\Delta E_{tot}$
is highest for $\beta$ values near zero. As seen in Fig. 5a, this range of $\beta$ corresponds to the regime of strong interaction (III). Energy
loss in this regime is maximal, both in the case of the ridged underside of the ice and in the case of smooth ice surfaces with the
same parameter $\beta$. When $\beta$ values increase, the dependence of energy loss on the $\mu$ and distance between the keels decreases.
$\beta = 0.8$ is on the boundary between regimes (II) and (I) (moderate ad weak) and the distance between the keels is no longer
significant.

In another limiting case, an ISW of elevation propagates over a corrugated bottom when the bottom element length was
much less than the ISW wavelength (Carr et al., 2010) a comparison with ISW propagated under an ensemble of ice keels
of horizontal scales greater than ISW length was not straightforward. In addition, Reynolds equations with turbulent closure
describe real-scale processes in the ocean, in contrast to laboratory scales in Carr et al. (2010). Unlike Carr et al. (2010), we
cannot describe in detail the instant spatial-temporal dynamics of high shear layer near the ice. However, Fig. 6 b shows wave-
induced currents over the keels, their interaction with the apex of the keels and a sequence of lee vortices formed as a result of
such interaction (see Fig. 6 b $T = 1$ h 35 m, $T = 1$ h 41 m). Similarly to Carr et al. (2010) the vortices developed after the main

wave passed over the keel (see Fig. 6 b at $T = 1$ h 44 m, $T = 1$ h 45 m) resulting in deformation of the overlying pycnocline and, in some instances, significant vertical mixing.

## 4 Conclusions

In this study, a numerical investigation of the transformation of ISW propagating from open water in a stratified sea into an ice covered region is carried out. We compared the transformation and energy loss of depression ISW under smooth ice surfaces, with the processes beneath ridged ice. It was shown that the transformation of depression ISWs under smooth ice cover is controlled by the blocking parameter $\beta$. Several regimes of ISW transformation at the ice-open water boundary were identified: (I) the weak interaction when the ISW transforms under ice cover without any instability; the energy losses are caused mainly due to viscous dissipation. It corresponds to values $\beta > 0.5$; (II) moderate interaction, which occurs when the waves become unstable under ice cover resulting in energy losses due to the turbulent mixing varying from $10\%$ to $20\%$; (III) strong interaction of ISW with the ice($\beta \simeq 0$) is the regime when the flow under the ice was supercritical and the values of energy loss were about $38\%$; and reflection regime (IV), when the depth of the ice cover is large enough to result in full reflection of the ISW. The ice's roughness has relatively little effect on energy conversions under ice cover.

The ISW transformation under ridged ice also depends on the blocking parameter $\beta$. For large keels ($\beta < 0$), more than $40\%$ of energy is lost on the first keel, while for relatively small keels ($\beta > 0.3$), the losses on the first keel are less than $6\%$. The energy losses in the flow around the ridges can be of the same order as for ice cover, in which the draft is commensurate with the amplitude of the keels. Energy losses due to all keels depend on the distance between them, which in turn depends on a keel quantity. These losses which is characterized by the parameter $\mu$ which is the ratio of keel depth to the distance between keels.

The energy loss processes of ISWs under ice deserve more in-depth studies to bridge with ISWs mixing and heat balance of polar oceans (Pinkel, 2005). The next step could be an explicit representation of heat and salt fluxes between the ice cover due to the ISW interaction with the ridged ice, e.g. following flux parametrization by (McPhee et al., 1987).

*Code availability.* The output files for all numerical experiments reported in the paper are available from the corresponding author.

*Author contributions.* VM designed the study, contributed to the visualization of the results, and wrote the manuscript with support from all authors. KT contributed to method development, simulation, data processing, and manuscript writing. ET contributed to the interpretation of the results and manuscript editing. All authors contributed to the article and approved the submitted version.

*Competing interests.* The authors declare that they have no conflict of interest.

*Acknowledgements.* This work has been supported by the Austrian Science Foundation (FWF) under projects P30887 and P31163 and the
European Union's Horizon 2020 research and innovation framework program (PolarRES, Grant Agreement 101003590). The authors would like to thank Kevin Lamb for his help in preparing the final version of the manuscript, useful comments and suggestions.

260

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
