# Peer review of "Transformation of internal solitary waves at the ice-open water boundary"

_EGUsphere, 2023_

## Author Comment (AC1)

**Reviewer#2.**

The authors are most grateful for your comments. We have followed your suggestions and revised the manuscript accordingly in many places. Please, find our responses below.

*This paper presents a numerical investigation of ISWs propagating under ice. A Reynolds averaged Navier-Stokes solver is utilised and both smooth and ridged ice is considered. The wave propagates from open water to under ice and two cases are focussed upon namely smooth ice and ridged ice. In the smooth ice case, a blocking parameter is shown to be the main control variable and flow dynamics in keeping with previous results by the first author and co-workers for an ISW of elevation over a step are seen. In the ridged case both the blocking parameter and a second parameter describing the ratio between keel depth and distance between keels are used to classify the flow.*

*The paper is original and interesting and I am supportive of publication subject to the minor remarks below.*

*The paper contains a lot of typographical and grammatical errors, these need to be fixed in advance of publication.*

**Answer**: The paper was checked to remove errors.

*The citation is not thorough enough. Key papers are cited but the authors often fail to compare their work with published literature.*

**Answer**: We added discussion and comparison of our simulation results with published experimental and numerical studies.

L. 102 "These results demonstrate a weak effect of free surface on ISW dynamics in considered cases which made it possible in this problem to replace the conditions on the free surface with conditions on the rigid lid. Note that in laboratory experiments (Carr et al., 2008; Luzzato-Fegiz and Helfrich, 2014) the influence of a free surface on the stability of waves with a trapped core was shown. This effect has been interpreted as the influence of surfactants essential in laboratory-scale processes, however, these Marangoni effects have a negligible impact on the interior of full-scale oceanic waves (Luzzatto-Fegiz and Helfrich, 2014)"

L. 212 "In the limiting case of the interaction of ISW with a single keel (Zhang et al., 2022b), the maximum energy dissipation was about 25% which is somewhat less than in our calculations, but we need to keep in mind the differences in the calculation parameters and turbulence parametrization. Zhang et al. (2022b) used constant eddy coefficients whereas in our study the model of turbulence was used where eddy coefficients vary in space and time."

L. 225 "In another limiting case ISW of elevation propagates over a corrugated bottom when the bottom element length was much less than the ISW wavelength (Carr et al., 2010) a comparison with ISW propagated under an ensemble of ice keels of horizontal scales greater than ISW wave length was not straightforward. In addition, Reynolds equations with turbulent closure describe real-scale processes in the ocean, in contrast to laboratory scales in (Carr et al., 2010). Unlike (Carr et al., 2010) we cannot describe in detail the instant spatial-temporal dynamics of high shear region near the ice. However, the Fig. 6b showed wave-induced currents over the keels, their interaction with the apex of the keels and a sequence of lee vortices

formed as a result of such interaction (see Fig. 6b T=1h 35 m, T=1h 41m). Similarly to (Carr et al., 2010) the vortices developed after the main wave passed over the keel (see Fig. 6b at T=1h 44 m, T=1h 45m) resulting in deformation of the overlying pycnocline and, in some instances, significant vertical mixing."

*Abstract second sentence – you refer to 'breaking IWs' at the edge. How do they break? Do they always break? Is there evidence for this? May be the word 'breaking' should be deleted?*

**Answer**:
Thank you for your comment. We have made changes in the text accordingly:

L. 2 "Transformation of the internal waves at the edge of the ice cover can essentially enhance the mixing and melting of ice in the Arctic Ocean and Antarctica."

*Abstract 3rd line – you talk about generation of ISWs, is this specific to polar oceans or in general?*

**Answer**: We have made changes in the text accordingly:

L. 3 "In the Polar Oceans the internal solitary waves (ISWs) are generated by various sources, including tidal currents over the bottom topography, the interaction of ice keels with tides, varying in time winds, vortices, and lee waves."

*Line 31. ISW shear, convective instabilities, and breaking on topographic inhomogeneities extract kinetic energy from ISWs for turbulence and **subsequent mixing increases the melting of ice**. Is the last part of this sentence true? If so can you give a suitable reference?*

**Answer**: We have changed the text to explain a sequence of processes:

L. 31 The transformation of an ISW under an ice keel can cause the advection of water below the ice layer, whereas ISW shear and convective instabilities result in turbulent mixing. The heat advection and turbulent flux both will contribute to the vertical heat flux and consequently the change in temperature under the sea ice and increase of melting (Zhang et al., 2022b).

*Line 50 – you say your wave goes from open water (with a free surface) to under-ice. Is this reflected in the numerical model or does the open water have a rigid lid in the numerical work? If so this should be made clear and potential differences with a free surface discussed.*

**Answer**: Thank you for your comment. We have refined the text as

L. 51 "In this study, a numerical investigation of the transformation of ISW propagating from ice-free water in the stratified sea under the edge of the ice cover is carried out to compare the depression ISW transformation and loss of energy on smooth ice surfaces, including those on the ice shelf, with the processes beneath the ridged underside of the ice."

To get around the difficulties associated with the numerical solution of the nonhydrostatic model equations in the presence of an ice layer, we considered the setting mirrored for the upper surface of the ocean, in which the ice layer was replaced by a step on the bottom. This approach requires using the rigid lid boundary condition at the ocean surface. Therefore, we

estimated the effect of free surface on the wave characteristics (L. 101 ). See answers on next comment.

*Line 100 – you have compared free slip and no slip and found little difference however it is known that the upper boundary condition can effect wave properties such as amplitude and stability at least on the lab scale (see e.g. Carr et al 2008 PoF, Luzzatto-Fegiz & Helfrich 2014 JFM). Why does it not matter here? Is it because surface tension effects aren't as important on your scale? Did you do any sensitivity test on the upper boundary condition?*

**Answer**: We have not compared free-slip and no-slip cases. In both model setups friction was taken into account only on the ice-water surface, whereas free-slip conditions were used at the rest of the boundaries (L. 85). The aim of tests with ISW the same amplitude propagating as a wave of depression and as a wave of elevation (see L. 97) was to estimate the effect of free surface on the wave characteristics for free-slip conditions. It was found that the difference in the horizontal velocity field between the two configurations of the model does not exceed 1% demonstrating a weak effect of free surface on ISW dynamics in considered cases. We have added text to clarify this conclusion

L. 101 "The tests aimed to estimate the effect of free surface on the wave characteristics for free-slip boundary conditions."

L. 102 "These results demonstrate a weak effect of free surface on ISW dynamics in considered cases which made it possible in this problem to replace the conditions on the free surface with conditions on the rigid lid. Note that in laboratory experiments (Carr et al., 2008; Luzzato-Fegiz and Helfrich, 2014) the influence of a free surface on the stability of waves with a trapped core was shown. This effect has been interpreted as the influence of surfactants essential in laboratory-scale processes, however, these Marangoni effects have a negligible impact on the interior of full-scale oceanic waves (Luzzatto-Fegiz and Helfrich, 2014)"

*Line 168 – you talk about reflected waves off the solid boundary step. Would you expect the same for real ice? Is there any way of assessing or inferring what will happen if the ice isn't solid for e.g in the MIZ when the ice is mushy?*

**Answer**: We assume that the ice layer is rigid and does not interact with ISWs (L. 65). The ISW interaction with floating ice plates and open water in MIZ is out of the scope of this study.

*Line 200 how does this statement compare with published papers on the generation of IWs by ice keels see e.g. Zhang et al 2022 J. Ocean Limnol, Zhang et al 2022 JGR:Oceans, M. McPhee & L. Kantha. 1989 J. Geophys. Res.*

**Answer:** The study of wave generation mechanisms is not discussed in this article. Investigation into the interaction of ISW with an ensemble of keels has not yet been carried out before our study. We added a discussion of the results of our simulations with two limiting cases: interaction ISW with a single keel (Zhang et al., 2022), and ISW propagation over a corrugated bottom when the bottom element length was much less than the ISW wavelength (Carr et al., 2010). The text has been added accordingly:

L.  212 "In the limiting case of the interaction of ISW with a single keel (Zhang et al., 2022b), the maximum energy dissipation was about 25% which is somewhat less than in our calculations, but we need to keep in mind the differences in the calculation parameters and turbulence parametrization. Zhang et al. (2022b) used constant eddy coefficients whereas in our study the model of turbulence was used where eddy coefficients vary in space and time."

L. 222  "If we assume that the tidal flow around the keels is the source of internal waves (Zhang et al., 2022), then we can conclude on the basis of our simulations that under conditions of strongly ridged ice, the waves excited by the tidal flow disperse in the vicinity of their formation."

L.  225 "In another limiting case ISW of elevation propagates over a corrugated bottom when the bottom element length was much less than the ISW wavelength (Carr et al., 2010) a comparison with ISW propagated under an ensemble of ice keels of horizontal scales greater than ISW wave length was not straightforward. In addition, Reynolds equations with turbulent closure describe real-scale processes in the ocean, in contrast to laboratory scales in (Carr et al., 2010). Unlike (Carr et al., 2010) we cannot describe in detail the instant spatial-temporal dynamics of high shear region near the ice. However, the Fig. 6b showed wave-induced currents over the keels, their interaction with the apex of the keels and a sequence of lee vortices formed as a result of such interaction (see Fig. 6b T=1h 35 m, T=1h 41m). Similarly to (Carr et al., 2010)  the vortices developed after the main wave passed over the keel (see Fig. 6b at T=1h 44 m, T=1h 45m) resulting in deformation of the overlying pycnocline and, in some instances, significant vertical mixing."

*Line 212 – the statement about ice roughness- is this in comparison to the blocking parameter?*

**Answer**: The text has been refined accordingly:

L. 190 The simulations showed a weak dependence of energy loss on the friction parameter CD (Fig. 5b)

*Line 222 - could the authors say more about this? How might this be represented within their numerical model for example?*

**Answer**: The text has been added accordingly:

L. 254 "The next step could be an explicit representation of heat and salt fluxes between the ice cover due to the ISW interaction with the ridged ice, e.g. following flux parametrization by McPhee et al.,(1987)."

---

## Author Comment (AC2)

**Reviewer#1.**

The authors are most grateful for your comments. We have followed your suggestions and revised the manuscript accordingly in many places. Please, find our responses below.

*This manuscript considers two-dimensional simulations of internal solitary waves propagating into a region with modelled ice cover. The ice cover is modelled as not moving, and as represented by a piecewise constant value (or perhaps as smoothed) of a drag parameter (which is varied to some degree). The former is sensible, while the latter is perhaps a necessary choice for the model employed. The manuscript is interesting, and the figures provide useful information. The text needs a thorough reading for technical English (if necessary I can provide a list of suggestions when the scientific review is completed). I feel that a version of this manuscript can appear in NPG, but there are some necessary changes/improvements. I enumerate these below, but as an overall comment I would say the results need to discuss the new results in terms of existing literature and the second set of experiments needs a more complete analysis and discussion.*

*I note that for many of the Yes/No questions the journal asks, the manuscript falls between a strict Yes or NO.*

**Answer**: We have revised and expanded the discussion of the results of the second series of simulations by adding a comparison with known laboratory experiments and numerical calculations (see responses to comments 3) and 7))

1) *Self-citation: Proofread to ensure that when a topic is introduced, e.g. shoaling of elevation, the references provided are more than just those of the authors (in particular for numerical studies). This is not just an issue of a longer bibliography. There are quite a few papers I would consider relevant listed, but they tend to appear as lists in the Introduction, and the opportunity to discuss the context of the numerical simulations in terms of these papers is missed.*

**Answer**: We expand our discussion of the results with comparisons with published works.

L. 103 "These results demonstrate a weak effect of free surface on ISW dynamics in considered cases which made it possible in this problem to replace the conditions on the free surface with conditions on the rigid lid. Note that in laboratory experiments (Carr et al., 2008; Luzzato-Fegiz and Helfrich, 2014) the influence of a free surface on the stability of waves with a trapped core was shown. This effect has been interpreted as the influence of surfactants essential in laboratory-scale processes, however, these Marangoni effects have a negligible impact on the interior of full-scale oceanic waves (Luzzatto-Fegiz and Helfrich, 2014)"

See also the answers to comment 7.

We have added literature:

McPhee, M. G., G. A. Maykut, and J. H. Morison: Dynamics and thermodynamics of the ice/upper ocean system in the marginal ice zone of the Greenland Sea, J. Geophys. Res., 92, 7017-7031, 1987.

Carr, M., Fructus, D., Grue, J., Jensen, A. & Davies, P. A.: Convectively induced shear

instability in large amplitude internal solitary waves. Phys. Fluids 20 (12), 126601, 2008.

Luzzatto-Fegiz P and Helfrich K: Laboratory experiments and simulations for solitary waves with trapped cores J. Fluid Mech. 757 354-380, 2014.

2) *Details of the numerical model. The basic idea of the top boundary condition is introduced, but right now an interested reader could not reproduce the results on their own. What needs to be done to implement the conditions? What complications result (e.g. in the pressure problem)? Can the drag coefficients really just be discontinuous?*

**Answer**: The text has been added to clarify numerical model details.

L. 86 "The pressure zero gradient boundary condition was imposed on all boundaries. At the corner of the underwater step, this condition is violated. However, numerical experiments for different resolutions have shown that this problem does not occur at simulated fields of velocity and density."

L. 111 "The quasi-z-level coordinate system (Maderich et al., 2012) was used to describe this step-like ice layer."

3) *The model resolution deserves comment. What can one expect to see/resolve (certainly I believe wave fissioning is accurately represented); what do we miss (I think the details of the high shear region near the ice cannot be accurately represented). The Carr et al corrugation paper gives details of the interaction with a no slip boundary layer, and hence provides an easy contrast.*

**Answer**: We applied the Reynolds averaged equations system closed by a simple subgrid model of turbulence to this real ocean scale problem. The standard boundary conditions for turbulent shear stress under rough ice surfaces were applied. Of course, this model cannot resolve small-scale structures in the turbulent flow. However, it can describe processes of ISW transformation and breaking. The comparison with the simulation of ISW transformation on a single ice keel (Zhang et al., 2022) showed very similar wave evolution despite differences in the parameterization of turbulent mixing (see answer to comment 7). The direct comparison of our simulations with laboratory and numerical experiments by Carr et al. (2010) is difficult since this work considers laboratory-scale processes and, in addition, the incident wave has a length much greater than the length of the corrugated bottom relief elements. Nevertheless, a certain similarity of flow processes was observed in both cases (see answer to comment 7).

4) *I'd prefer "smoothed step" to "step". It would also help to state for the reader how many points there are across the changing part of the tang-based step.*

**Answer**: The first series of experiments was performed using a quasi-z-coordinate which allowed us to describe a step-like ice layer without any smoothing. The text has been added:

L. 111 "The quasi-z-level coordinate system (Maderich et al., 2012) was used to describe this step-like ice layer."

In the second series of experiments, the sigma coordinate system was used to accurately describe flow around the keels placed under a relatively thin ice layer.

L.  118 In the second series of experiments (see Table 2), 12 runs (K1-K12) were performed using a sigma-system of coordinates, which allowed for accurately describing flow around the keel.

5) *It would be good to indicate the integration region for the energetics calculations on the appropriate panel of Fig 4.  Similarly, the spacing of the equations in the system 6 could be improved (perhaps this is due to them lying at the bottom of the page, and they will likely move in a final version of the manuscript).*

**Answer**: Thank you for the suggestion. We have made changes in the fig 4 and corresponding caption.

[Figure]

Figure 4. The snapshots of the density field for incident ISW waves with an amplitude of 15 m passing under the ice cover with different drafts. The integration region for the energetics calculations between $X_{l1}$ and $X_R$ is shown.

6) *Presumably the g in equation (8) is a reduced gravity?  Otherwise I cannot see how a supercritical regime is reached.*

**Answer**: Thank you!  We corrected g to g'.

L  178  $$Fr^2 = \frac{U_1^2}{g'h_1(x)} + \frac{U_2^2}{g'h_2(x)}$$

where U1 and U2 are the layer-averaged velocities in each layer, $g' = g\Delta\rho/\rho_0$, where g is the gravity acceleration, $\Delta\rho$ and $\rho_0$ are the density difference between upper and lower layers and undisturbed density of fluid, respectively, g is the gravity acceleration.

7) *I found the second set of experiments, the ice keels, to be a bit tougher to digest, likely due to its brevity.  I have questions about the way the boundary layer is parametrized (wouldn't there be more drag over the downstream slope where Carr at all predicted "local hydraulic"*

*phenomena?). The discussion of Fig 7 seems incomplete (what are the curves shown, what are the details of what is presumably a fitting process?).*

**Answer**: We extended the discussion of the second series of experiments including Fig. 7 and compared the results of our simulations with two limiting cases: interaction ISW with a single keel (Zhang et al., 2022), and ISW propagation over a corrugated bottom when the bottom element length was much less than the ISW wavelength (Carr et al. 2010). The text has been added accordingly:

L. 212 "In the limiting case of the interaction of ISW with a single keel (Zhang et al., 2022b), the maximum energy dissipation was about 25% which is somewhat less than in our calculations, but we need to keep in mind the differences in the calculation parameters and turbulence parametrization. Zhang et al. (2022b) used constant eddy coefficients whereas in our study the model of turbulence was used where eddy coefficients vary in space and time."

L. 215 "To characterize the dependence of $\Delta E_{tot}$ on keel height and distance between keels we introduced parameter $\mu = (h_{ice} + h_k)/L_k$. As seen in Fig. 7 this dependence can be approximated by logarithmic curves. The energy loss $\Delta E_{tot}$ increases with the decrease of distane between keels or an increase of keel height. The level of $\Delta E_{tot}$ is highest for β values near zero. As seen in Fig. 5a, this range of β corresponds to the regime of strong interaction (III). Energy loss in this regime is maximal, both in the case of the ridged underside of the ice and in the case of smooth ice surfaces with the same parameter β. When β values increase, the dependence of energy loss on the μ and distance between the keels decreases. For β=0.8 is on the boundary between regime (II) moderate and (I) weak interaction distance between the keels is no longer significant."

L. 225 "In another limiting case ISW of elevation propagates over a corrugated bottom when the bottom element length was much less than the ISW wavelength (Carr et al., 2010) a comparison with ISW propagated under an ensemble of ice keels of horizontal scales greater than ISW wave length was not straightforward. In addition, Reynolds equations with turbulent closure describe real-scale processes in the ocean, in contrast to laboratory scales in (Carr et al., 2010). Unlike (Carr et al., 2010) we cannot describe in detail the instant spatial-temporal dynamics of high shear region near the ice. However, the Fig. 6b showed wave-induced currents over the keels, their interaction with the apex of the keels and a sequence of lee vortices formed as a result of such interaction (see Fig. 6b T=1h 35 m, T=1h 41m). Similarly to (Carr et al., 2010) the vortices developed after the main wave passed over the keel (see Fig. 6b at T=1h 44 m, T=1h 45m) resulting in deformation of the overlying pycnocline and, in some instances, significant vertical mixing."

8) *I agree with the comments in line 220. At the same time, I think there have been simulations in the literature of related heat-salt phenomena. Tt least to point to these as a start of relevant studies.*

**Answer**: The text was added accordingly:

L. 254 "The next step could be an explicit representation of heat and salt fluxes between the ice cover due to the ISW interaction with the ridged ice, e.g. following flux parametrization by McPhee et al., (1987)."

---

## Referee Report (RR1)

The authors have done a good job in responding to the various reviewer/editor comments. I think the manuscript is ready to be published.

---

## Author Response (AR2)

**Answers to the Editor's comments**

The authors are most grateful for your comments. We have followed your suggestions and revised the manuscript accordingly in many places. Please, find our responses below.

*1. The title of the paper is somewhat misleading as the vast majority of the simulations considered in the paper are on the transformation at the ice-edge boundary.*

**Answer**: Thank you for the suggestion. We changed the title to "Transformation of internal solitary waves at the ice-open water boundary"

*2. Following up on the previous point, dynamically an ISW propagating past the ice-edge boundary is equivalent to the setup in previous work which considered ISWs propagating past a step at the bottom. It should be made clear what is new here for this set of simulations. What is different about this set of simulations and what are the new conclusions? For example Figure 2 in Talipova et al. 2013 and the current manuscript are very similar.*

**Answer**: In this study, we used the Reynolds equations and a turbulence model to describe field-scale processes, in contrast to Talipova et al (2013), which considered laboratory-scale processes and used the Navier-Stokes equations. Therefore, there is not a direct correspondence between our calculations and Talipova et al (2013) in Fig. 2, although they are qualitatively close. Text was changed accordingly:

L 189 "The differences in values of the energy losses from (Talipova et al., 2013) and from the present investigation can be explained by the fact that the field scale problem was studied in this work using the Reynolds averaged equation, while in Talipova et al. (2013) the propagation of ISWs in a laboratory-scale computational domain was studied by using the Navier-Stokes equations."

*3. On page 4 it is stated that a zero pressure gradient boundary condition is used. This can not be correct as a non-zero pressure gradient exists at the boundaries above and below an ISW. This, after all, is what accelerates the fluid near the boundaries.*

**Answer**: Thank you for the comment. The text was changed accordingly:

L 89 "The Neumann-type boundary condition for the nonhydrostatic pressure component was used at the solid boundaries. At the free surface and open boundaries, this component was set zero (Maderich et al., 2012)."

*4. It is mentioned that this work considers the 'real scale' situation (e.g., page 11). What exactly is meant by this? The only way it can be field scale (I assume this is what is meant) is if the Reynolds number is appropriately large. The Reynolds number is never mentioned. Neither is the value of the viscosity used in these simulations.*

**Answer**: Thank you for the suggestion. The field-scale Reynolds number based on ISW amplitude and velocity is an order $10^7$ which forces the use of Reynolds equations closed by a turbulence model. We changed 'real scale' to 'field scale' and added text:

L. 192 "The eddy viscosity and diffusivity calculated from the turbulence model (Siegel, Domaradzky, 1994) vary in space and time with characteristic values $10^{-4}$-$10^{-3}$m$^2$s$^{-1}$."

*5. For the simulations of an ISW propagating beneath multiple ice keels the results are discussed in terms of the distance between the keels. Wouldn't the primary dependence be on the number of ice keels?*

**Answer**: Thank you for the suggestion. For a finite length of the ice layer $L_{ice}$, as in the calculations under consideration, the distance between the keels $L_k$ is expressed through $L_{ice}$ and the number of keels n. We indicated this in the text

L.131 "In turn, for a finite length of the ice layer $L_{ice}$ the distance between keels depend on keel number n as $L_k = L_{ice}/n$".

When $L_{ice}$ asymptotically increases then n also increases and the only governing parameter is $L_k$.

*Also, I have provided a marked up copy to help with English corrections. The list is not complete but it should be a good start.*

**Answer**: Thank you very much. We corrected the text in many places accordingly.

---

## Author Response (AR3)

The authors are most grateful for your comments. We have followed your suggestions and made the changes that were indicated. Please, find our responses below.

**Reviewer#1.**

The manuscript is ready to be published, but should get a read over by technical staff for copy-editing and "smoothing" of the technical english

**Answer**: We have made some changes to the text. Typos, extra spaces, and punctuation.

**Reviewer#2.**

I find the new title more misleading than the old ! The numerical scheme does not replicate an ice-open water boundary as it is fixed/solid boundary everywhere. May be using the words ice-edge would be more appropriate ?

**Answer**: The title was changed accordingly:

Transformation of internal solitary waves at the edge of the ice cover.